# Associations between the microbiome and immune responses to an adenovirus-based HIV-1 candidate vaccine are distinct between African and US cohorts

Yuhao Li,[1] Daniel J. Stieh,[2] Lindsay Droit,[3] Andrew HyoungJin Kim,[1] Rachel Rodgers,[3] Kathie A. Mihindukulasuriya,[3] Leran Wang,[3] Maria G. Pau,[2] Olive Yuan,[4] Herbert W. Virgin,[3,5] Dan H. Barouch,[6,7] Megan T. Baldridge,[1,8] Scott A. Handley[3,8]

**ABSTRACT**  Optimization of prophylactic vaccine regimens to elicit strong, long-lasting immunity is an urgent need highlighted by the COVID-19 pandemic. Stronger vaccine immunogenicity is frequently reported in individuals living in high-income countries compared to individuals living in low- and middle-income countries. While numerous host genetic and immune factors may influence vaccine responses, geographic restrictions to vaccine effectiveness may also be influenced by the intestinal microbiota, which modulates host immune systems. However, the potential role of the gut microbiota on responses to HIV-1 vaccines has not yet been explored. We analyzed the bacteriome by targeted 16S sequencing and the virome by virus-like particle sequencing of 154 fecal samples collected from healthy individuals in Uganda, Rwanda, and the United States early (week 2) and late (week 26) after vaccination with multivalent adenovirus serotype 26 (Ad26)-vectored mosaic HIV-1 vaccines. Vaccination did not affect the enteric bacteriome or virome regardless of geographic location. However, geography was the major driver of microbiota differences within this cohort. Differences in overall bacterial and viral diversity and in specific microbial taxa, including *Bacteroidota* and *Bacillota*, between participants from the United States and East African countries correlated with differential immune responses, including specific antibody titers, antibody functionality, and cellular immune responses to vaccination regimens. These findings support the microbiota as a putative modifier of vaccine immunogenicity.

**IMPORTANCE** Our research examined how gut bacteria might influence vaccine effectiveness in different parts of the world. We studied adults from the United States, Rwanda, and Uganda who received an experimental HIV vaccine. We found that participants from East Africa had more diverse gut bacteria than those from the United States, but their immune responses to the vaccine were weaker. This is the first study to directly show this relationship between higher gut bacterial diversity and reduced vaccine effectiveness in the same group of people. We also identified specific types of bacteria that were linked to either stronger or weaker immune responses. These findings are particularly relevant now as we use vaccines globally to fight diseases like COVID-19, as they suggest that regional differences in gut bacteria *Bacteroidota* and *Bacillota* might help explain why vaccines work better in some places than others. This could inform how we design and test future vaccines.

**KEYWORDS**    microbiome, virome, HIV, vaccine, VLP

According to the United Nations Programme on HIV/AIDS (UNAIDS), at the end of 2023, there were approximately 39.9 million people living with HIV-1 worldwide, with 11.3 million people newly infected and 630,000 patients dying from AIDS-related

Address correspondence to Scott A. Handley, shandley@wustl.edu, Megan T. Baldridge, mbaldridge@wustl.edu, or Dan H. Barouch, dbarouch@bidmc.harvard.edu.

D.J.S. and M.G.P. are employees and shareholders of Janssen Vaccines and Prevention B.V., the regulatory sponsor of the study, which is a subsidiary of Johnson & Johnson.

See the funding table on p. 20.

illnesses (1). The vast majority of HIV-infected patients are living in low- and middle-income countries (LMICs), and despite the availability of proven pre-exposure prophylaxis and combined antiretroviral therapy to combat HIV-1 infection, only 77% (62%–90% of adults aged 15 years and older) living with HIV have access to treatment (1). Currently, a cure for HIV-1 remains elusive, making the development of a safe, effective, and affordable HIV-1 prophylactic vaccine a top priority. In the past several decades, a series of vaccine candidates leveraging different strategies have been tested in Phase I/II/III clinical trials in both high-income countries (HICs) and LMICs (2–6). Despite rigorous pre-clinical research, most of these vaccine candidates have shown no or limited efficacy in preventing HIV infection in large-scale clinical trials (7).

The design of an effective prophylactic HIV-1 vaccine demands overcoming numerous obstacles. HIV-1 targets and impairs major host immune cells, integrates into the host cell genome, and rapidly establishes latency in undefined reservoirs (6). Another major challenge for vaccine development is the enormous global genetic diversity of HIV-1. HIV-1 consists of four phylogenetically distinct groups (O, P, N, and M), with group M (main) globally dominating (8). Group M has evolved into nine subtypes (A–D, F–H, J, and K) (9). The most common variant of HIV-1 is subtype C (46.6% infection rate globally), followed by subtype B (12.1%) and subtype A (10.3%) (10). HIV-1 subtype B is prevalent in Western and Central Europe, Latin and North America, and Oceania, while subtype A is a common variant in East Africa, contributing to 53.4% of all infections in the region. HIV-1 subtype C is mostly detected in low-income regions of India, Ethiopia, and Southern Africa and is increasing in prevalence in Eastern Europe and Eastern Africa (11). The high rates of viral mutation and recombination during HIV-1 replication limit the potency and coverage of both humoral and cellular immune responses induced by selective vaccine regimens. To overcome this limitation, multivalent adenovirus serotype 26 (Ad26)-vectored mosaic HIV-1 vaccines, which elicit cross-clade immune responses, have been developed (12–15). These mosaic immunogens are *in silico*-derived, full-length recombinants of natural HIV-1 proteins, optimized to maximize potential T-cell epitopes (PTEs) and permit natural expression, antigen processing, and presentation (16, 17). Among several vaccine combinations, Ad26 vector expressing computationally optimized mosaic HIV-1 envelope (Env)/Gag/Pol immunogens has been shown to induce robust and specific immune responses in both peripheral blood and colorectal mucosa, indicating its promise as a vaccine candidate (18–21). However, analysis of a phase 2b proof-of-concept study in sub-Saharan Africa (HVTN 705/HPX2008) suggests that Ad26 mosaic HIV-1 vaccination offers limited protection against HIV-1 infection in women aged 18–35 (21), and an additional phase 3 trial in men who have sex with men and transgender populations in the Americas and Europe (HVTN 706/HPX3002) was discontinued (22) after a planned, interim review revealed that it was not effective at preventing the acquisition of HIV-1 (22).

As optimization of prophylactic vaccine regimens to elicit strong and long-lasting immunity continues, another important consideration has begun to emerge, specifically that there is high variation in immune responses to vaccination associated with an individual's country of origin (23). Compared to individuals living in LMICs, stronger vaccine immunogenicity has frequently been reported in individuals living in HICs (23), with differential responses reported for influenza, yellow fever, and Ebola candidate vaccines (24–26). Similarly, in an Ad26/Ad35 vector-based HIV-1 vaccine trial, participants from East Africa exhibited diminished T cell immune responses compared to participants from South Africa or the United States (27). While numerous host genetic and immune factors have the potential to influence vaccine responses, the geographic restrictions to vaccine effectiveness may also be influenced by intestinal microbiota. The intestinal microbiota is intimately involved in both the development of the immune system and the regulation of immune responses (23, 28, 29). Understanding how the intestinal microbiota specifically influences responses to adenovirus-vectored vaccines is important for addressing the observed geographic disparities in vaccine immunogenicity between HIC and LMIC populations observed in this study. Further evidence

of a role for the microbiome influencing vaccine effectiveness comes from antibiotic administration studies. Antibiotic alteration of the bacterial microbiota has been shown to modulate responses to rotavirus and influenza vaccines in human adults (30, 31), indicating that robust disruption of the microbiota can drive dramatic changes in vaccine efficacy. In mouse studies, influenza vaccine responses are inversely associated with microbiota complexity and natural pathogen exposure history (32), supporting a role for microbiome influence on these responses. However, despite evidence that responses to parenteral vaccines are diminished in individuals from LMICs (24–27), a limited number of studies to date have investigated the link between immune responses to intramuscular vaccines and the intestinal microbiome (33–36).

Vertebrate viral co-infections have also been implicated as potentially influencing vaccine responses (37). For example, recently acquired enterovirus infections have been associated with ineffective immune responses to oral poliovirus and rotavirus vaccines (38–40). The role of enteric virus coinfection is particularly important in the context of HIV-1 infection and vaccination, as enteropathogenic viruses are associated with the development of AIDS in simian immunodeficiency virus (SIV)-infected macaques and HIV-1-infected individuals with low (fewer than 200 cells/mL) CD4$^+$ T cells (41, 42). In addition, SIV-infection mediated changes in the enteric virome are prevented by vaccination (43). Taken together, these studies suggest an important axis connecting enteric virus co-infection to vaccine-preventable pathophysiology associated with SIV and HIV-1 infection.

Thus, there is emerging evidence suggesting important interactions between the bacterial and viral microbiota and vaccine responses, but the potential role of the gut microbiota on responses to HIV-1 vaccines has not yet been explored. Additionally, no analysis to date has explored the associations between the enteric microbiome and HIV-1 clade-specific immune responses in the context of regional HIV-1 strain diversity, as dominant circulating HIV-1 clades differ by country (44). Here, we leveraged microbiota samples collected early and late in the vaccination time course from a multicenter, randomized, double-blind, placebo-controlled phase I/IIa trial evaluating an Ad26-mosaic-based HIV-1 vaccine candidate (APPROACH) (21). We analyzed the enteric bacteriome and virome composition of a subset of the healthy, HIV-1-uninfected participants recruited from two regions with distinct HIV-1 clade prevalence: East Africa, including Rwanda and Uganda, which has a high prevalence of HIV-1 subtype A, and the United States, with a high prevalence of subtype B. Our analysis characterized bacterial and viral microbiome composition in both regions and identified regional microbial associations with specific vaccine responses.

This study aims to investigate whether geographic differences in intestinal microbiota influence immune responses to Ad26-vectored HIV vaccines by analyzing bacterial and viral populations in fecal samples from participants in the United States, Rwanda, and Uganda. Specifically, we sought to determine if variations in gut microbial diversity and composition correlate with differences in vaccine immunogenicity, including antibody titers, antibody functionality, and cellular immune responses across these geographically distinct populations.

## MATERIALS AND METHODS

### Subjects and samples

Subjects and samples were from the phase I/IIa clinical trial APPROACH (https://clinicaltrials.gov/study/NCT02315703), which has been previously published (21). All participants provided written informed consent as previously described (21). In brief, healthy, HIV-1-uninfected participants were recruited and randomly assigned to one of eight study groups: Ad26/Ad26 plus high-dose gp140, Ad26/Ad26 plus low-dose gp140, Ad26/Ad26, Ad26/MVA plus high-dose gp140, Ad26/MVA plus low-dose gp140, Ad26/MVA, Ad26/high-dose gp140, and placebo. All interventions were administered intramuscularly. The vaccine group participants were primed with Ad26.Mos.HIV ($5 \times 10^{10}$

viral particles per 0.5 mL) at weeks 0 and 12 and subsequently boosted with one of the following combinations: Ad26.Mos.HIV with or without high-dose gp140 protein (250 µg) or low-dose gp140 protein (50 µg), MVA-mosaic ($10^8$ plaque-forming units per 0.5 mL) with or without high-dose or low-dose gp140 protein, or high-dose gp140 protein alone at weeks 24 and 48. Placebo group participants received 0.9% saline at weeks 0, 12, 24, and 48. Blood and fecal samples were collected at early (week 2) and late (week 26) time points.

Human immunologic assays for the APPROACH trial were performed as previously described (21, 45–47). Vaccine response was defined as titer > threshold (if baseline is less than threshold or is missing); otherwise, as a titer with a threefold increase from baseline (if baseline is greater than or equal to threshold). Humoral immune responses to each vaccine group were measured by ELISA and antibody-dependent phagocytosis (ADCP) assays, and cellular immune response was measured by enzyme-linked immunospot (ELISpot) assay. Vaccine response was defined as geometric mean titers of greater than 200 spot-forming cells to one or more HIV antigens (ELISpot). The gp140 antigens used were WT_A_92UG037 (clade A), WT_B_1990A (clade B), WT_C_CONC (consensus clade C), VaccineTake_C_C97ZA (clade C), and VaccineTake_Mos1 (mosaic).

## 16S rRNA amplicon sequencing and analysis

DNA was extracted from 154 fecal pellets using a phenol:chloroform method for 16S rRNA gene sequencing. Primer selection and PCRs were performed as described previously (48). Briefly, each sample was amplified in triplicate with Golay-barcoded primers specific to the V4 region (F515/R806), combined, and confirmed by gel electrophoresis. PCR reactions contained 18.8 µL RNase/DNase-free water, 2.5 µL 10× High Fidelity PCR Buffer (Invitrogen), 0.5 µL 10 mM dNTPs, 1 µL 50 mM $MgSO_4$, 0.5 µL each of the forward and reverse primers (10 µM final concentration), 0.1 µL Platinum High Fidelity Taq (Invitrogen), and 1.0 µL genomic DNA. Reactions were held at 94°C for 2 min to denature the DNA, with amplification proceeding for 26 cycles at 94°C for 15 s, 50°C for 30 s, and 68°C for 30 s; a final extension of 2 min at 68°C was added to ensure complete amplification. Amplicons were pooled and purified with 0.6× Agencourt Ampure XP beads (Beckman-Coulter) according to the manufacturer's instructions. The final pooled samples, along with aliquots of the three sequencing primers, were sent to the DNA Sequencing Innovation Lab (Washington University School of Medicine) for sequencing by the $2 \times 250$ bp protocol with the Illumina MiSeq platform.

Read quality control and the resolution of amplicon sequence variants (ASVs) were performed with the dada2 R package (49). ASVs that were not assigned to the kingdom Bacteria were removed. The remaining reads were assigned taxonomy using the Ribosomal Database Project (RDP trainset 16/release 11.5) 16S rRNA gene sequence database (50). Ecological analyses, including alpha-diversity (richness and Faith's phylogenetic diversity [PD]) and beta-diversity analyses (weighted UniFrac distances), were performed using PhyloSeq and additional R packages (51), and differentially abundant taxa between groups were identified by performing pairwise comparisons using DESeq2 (52). All 16S sequences were uploaded to the European Nucleotide Archive under project PRJEB48706.

## Virome sequencing and analysis

Virus-like particles (VLPs) were prepared from 152 stool samples as previously described (53, 54). Approximately 200 mg of chipped fecal specimens was resuspended in phosphate-buffered saline and clarified by filtration through a 0.45-µm-pore-size filters until clarified. Clarified samples are subsequently treated with lysozyme to liberate bacterial nucleic acid, followed by DNase treatment to remove non-encapsidated viral nucleic acid. Total nucleic acid (both RNA and DNA) was extracted on a COBAS AmpliPrep instrument (Roche) according to the manufacturer's recommendations. Purified total nucleic acid was reverse-transcribed and PCR amplified using barcoded primers consisting of a base-balanced 16-nucleotide-specific sequence (Table S1) and used

for NEBNext library construction (New England BioLabs). Libraries were multiplexed (12 samples per flow cell) on an Illumina MiSeq instrument (Washington University Center for Genome Sciences) using a 2 × 250 bp paired-end protocol. All unprocessed virome sequences were uploaded to the European Nucleotide Archive under project PRJEB48706.

Unprocessed paired-end reads were processed through a multistage quality-control procedure to remove primers and adapters, human and other contaminant and low-quality sequence data (Table S2). High-quality paired-end reads, kmer-normalized using bbnorm (target depth = 20 and minimum depth = 2), were assembled for each sample using MEGAHIT (default settings) (55, 56). A study-wide contig dictionary was generated by supplying MEGAHIT output contigs to Flye (--meta), essentially assembling the assemblies (57, 58). Contigs were assigned taxonomy using iterative MMseqs2 searching against both nucleotide and protein viral databases. The abundance of each contig per sample was determined by mapping individual sample reads to the Flye-generated contig dictionary using Kallisto (59) (further details in Table S2).

## Statistical analysis

Details of specific statistical analyses are provided in the figure legends and Results. All analyses were performed using R version 4.1.1.

## RESULTS

### Properties of sample collection and 16S rRNA amplicon and virome sequences

Between 24 February 2015 and 16 October 2015, an initial phase I/IIa vaccine trial of the Ad26 mosaic gp140 regimen with a trivalent vector combination (Ad26.Mos.HIV) encoding a single Mosaic Env antigen and bivalent Mosaic Gag-Pol antigens was tested on participants, including those from the United States, Uganda, and Rwanda (21). From the 279 participants at these sites who were randomly assigned to receive at least one dose of test vaccine or placebo, 153 total fecal samples were collected at either an early (week 2) or late (week 26) time point in the vaccination time course (Table 1). Of 153 fecal samples, 70 (46%) were collected from the United States, 31 (20%) from Uganda, and 52 (34%) from Rwanda. Eighty-four of these samples were collected from the same individual at two collection time points (from 29 participants in the United States and 13 participants in Uganda), while all samples from Rwanda were collected at the late time point. To address outstanding questions related to putative interactions between the microbiota and HIV-1 vaccine responses, we analyzed the bacterial microbiome using 16S rRNA (V4) surveys and the stool virome using shotgun sequencing. Both RNA (converted to cDNA) and DNA viruses were analyzed from VLP preparations. Some samples failed to generate libraries for virome analysis, but the overall paired success rate was greater than 90% at each geographic region and time point (Table 1).

We obtained 16S rRNA amplicon data for a total of 70 and 31 samples, respectively, from the United States and Uganda (Table 1). 16S rRNA amplicon data were only available for a single placebo sample from Uganda at each early and late time point. The Ugandan vaccine group had 13 early samples (average of 1.9 samples per vaccine arm; SD: 1.21) and 16 late samples (average of 2.28 samples per vaccine arm; SD: 1.38). 16S rRNA amplicon data were generated from a larger cohort of the US samples at both early ($n = 34$) and late ($n = 36$) post-vaccination time points. This included four placebo samples at both early and late post-vaccination time points and from an average of 4.29 (SD: 2.06) early and 4.57 (SD: 2.64) late samples per vaccine arm. Sampling from Rwanda was limited to the late post-vaccination time point, but it had the most extensive sampling of any region and time point, with a total of 52 samples. Therefore, Rwandan samples were only used in non-longitudinal analyses. 16S rRNA amplicon data were obtained from 6 placebo samples and a total of 46 post-vaccination samples with an average of 6.57 (SD: 1.27) samples per vaccine arm. In total, 16S rRNA

**TABLE 1** Summary of fecal samples collected and analyzed in this study[a]

| Sample collection group | Early time point (16S/virome) | | Late time point (16S/virome) | | | Total |
|---|---|---|---|---|---|---|
| | US | UG | US | UG | RW | |
| Placebo | 4/3 | 1/1 | 4/4 | 1/1 | 6/6 | 16/15 |
| Vaccine | | | | | | |
| Ad26/Ad26 plus high-dose gp140 | 6/6 | 1/1 | 7/7 | 1/1 | 8/8 | 23/23 |
| Ad26/Ad26 plus low-dose gp140 | 2/2 | 1/1 | 1/1 | 1/0 | 7/7 | 12/11 |
| Ad26/Ad26 | 6/6 | 3/3 | 5/5 | 4/4 | 8/8 | 26/26 |
| Ad26/MVA plus high-dose gp140 | 3/3 | 1/1 | 2/2 | 1/1 | 6/6 | 13/13 |
| Ad26/MVA plus low-dose gp140 | 2/2 | 1/1 | 3/3 | 2/2 | 7/7 | 15/15 |
| Ad26/MVA | 4/3 | 4/3 | 6/4 | 4/4 | 5/5 | 23/19 |
| Ad26 plus high-dose gp140 | 7/7 | 2/2 | 8/8 | 3/3 | 5/5 | 25/25 |
| Total | 34/32 | 14/13 | 34/32 | 14/13 | 63/63 | 153/147 |

[a]Fecal samples collected from healthy, HIV-1-uninfected participants from the United States (US), Rwanda (RW), and Uganda (UG) in the APPROACH clinical study. Early and late post-vaccination time points are at week 2 and week 26 after the first vaccination, respectively. Ad26, adenovirus serotype 26. MVA, modified vaccinia Ankara.Gp140, aluminum phosphate-adjuvanted clade C gp140 Env protein in high dose (250 µg) or low dose (50 µg).

amplicon sequencing data were obtained from all 153 samples for a total of 7,633,594 (average of 49,892 reads per sample; SD: 26,460) reads. Resolution of ASVs using DADA2 yielded 3,671 distinct ASVs, which were taxonomically assigned by MMSeqs2. Filtering for presence in ≥3% of samples yielded 3,135 ASVs. DESeq2's internal filtering further refined this to 1,940 ASVs for differential abundance testing (Table S4).

Virome sequencing followed similar sampling properties as the 16S rRNA amplicon surveys described above (Table 1). The failure rate to obtain virome data was slightly higher in comparison to 16S rRNA data acquisition, resulting in six fewer virome samples, four fewer from the United States (two from both early and late time points) and two fewer from Uganda (one from both early and late time points). For the Rwandan cohort, virome sequencing data were successfully generated for every sample that also underwent 16S rRNA sequencing, with no virome-specific sample failures. Virome sequencing generated a total of 134,551,224 paired-end reads from 152 samples, with an average sequencing depth of 885,205 sequences per sample (SD: 502,547). Virome sequences were processed through a quality control pipeline to remove non-biological (primers and adapters) and irrelevant human sequences prior to metagenomic assembly. Quality control removed an average of 64,116 read pairs per sample. Per sample metagenomic assembly resulted in 194,637 total contigs longer than 1,000 kilobases in length, with an average of 1,280 contigs per sample (SD: 1,398). A unique set of 12,171 contigs was generated using Flye with the meta flag (Table S2). Per sample contig abundance was estimated as sequences per kilobase million (SPM) to normalize for contig length and per sample sequencing depth. SPM values ranged from 1 to 1,780, with an average SPM of 75.24 contigs across the entire set of samples.

Due to insufficient samples in individual vaccine arms for robust statistical analysis, we analyzed all vaccinated individuals together as a single "vaccinated" group.

## Immune responses to Ad26-based HIV-1 vaccination regimens vary between geographic regions

The immune responses to vaccination regimens are summarized in Fig. 1A and Table S3 (21). All vaccine regimens were highly immunogenic. Binding antibody responses to autologous Env clade A (92UG037.1), clade B (1990a), clade C (consensus clade C and Con C), and mosaic (Mos1) gp140 were detected in the vast majority of vaccinees evaluated at late time points (100%; 95% CI, 93%–100%). ELISA titers were generally higher in the samples from the United States compared to participants from Uganda and Rwanda (Fig. S1A). Titers to clade A antigen were higher in participants from the United States than from both East African nations (Dunn's test, adjusted $P < 0.05$), while titers

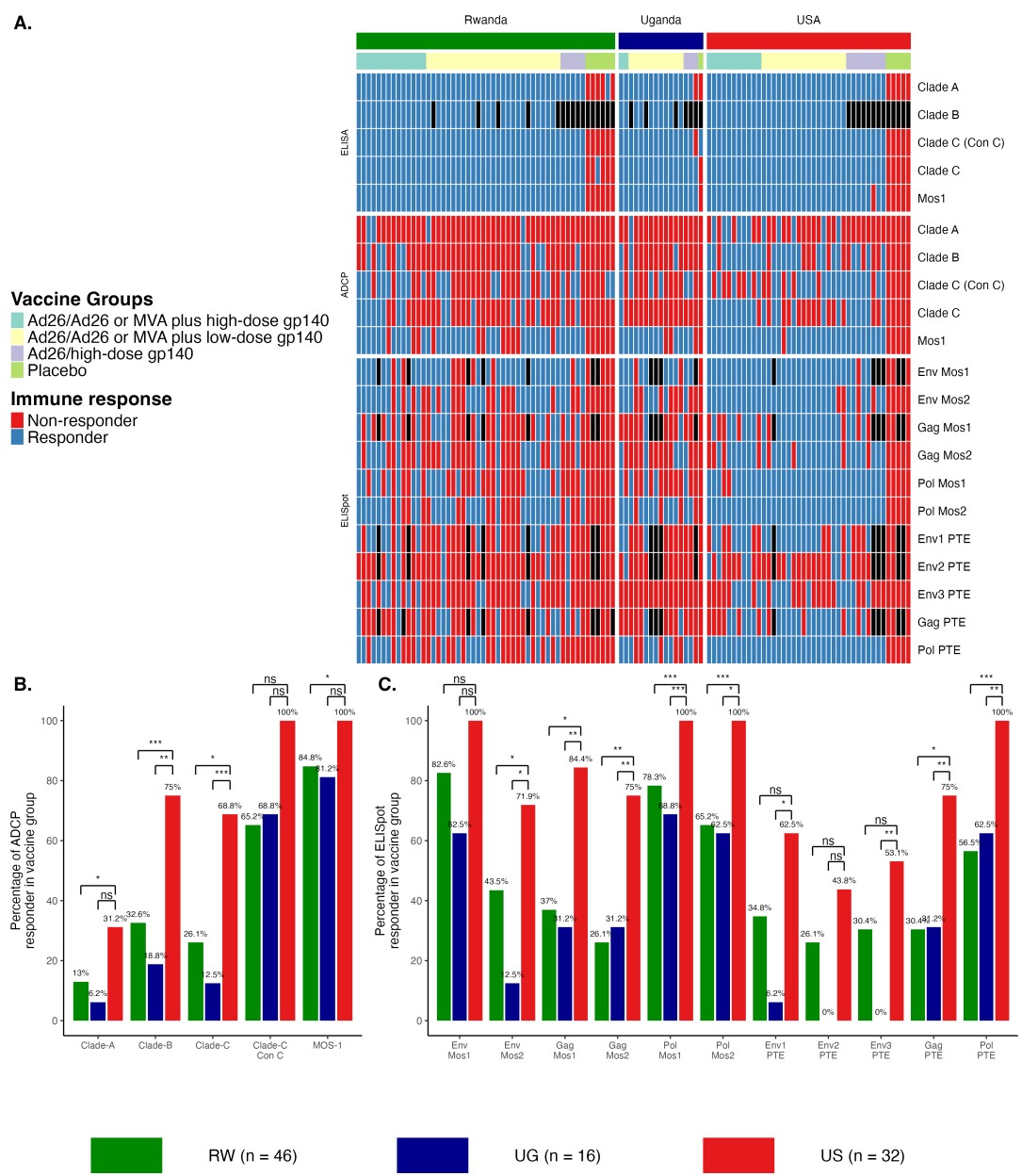

**FIG 1** Immune responses vary between healthy individuals from different geographic regions. (A) Heatmap of immune responses, indicating individual APPROACH study participants' response or non-response for all ELISA, ADCP, and ELISpot assays for the indicated clade-specific antigens or peptide pools. Percentage of responders for HIV-1 clade-specific (B) ADCP or (C) ELISpot responses for vaccinated participants from Rwanda (RW), Uganda (UG), and the United States (US). Vaccine response was defined as a value more than the threshold (if baseline is less than the threshold or is missing); otherwise, it was defined as a value with a three-time increase from baseline (if baseline is greater than or equal to the threshold) for Env-specific gp140 antibody-dependent phagocytic cells or antibody titers (ADCP and ELISA). Vaccine response was defined as geometric mean titers of greater than 200 spot-forming cells to one or more HIV antigens (ELISpot) as described previously (21, 22, 45–47). Statistical significance was determined using Fisher's exact test adjusted for multiple comparisons using the Benjamini-Hochberg procedure. ****$P < 0.0001$; ***$P < 0.001$; **$P < 0.01$; *$P < 0.05$; and ns, not significant.

to clade B, C, and Mos1 antigens were significantly higher in participants from both the United States and Rwanda than from Uganda (Dunn's test, adjusted $P < 0.05$). ELISA titers to consensus clade C (Con C) antigen were higher only in participants from the United States in comparison to those from Uganda (Dunn's test, adjusted $P < 0.05$).

ADCP scores, an evaluation of antibody functionality, were consistently higher in participants living in the United States when compared to participants living in either Uganda or Rwanda for all antigens (Fig. S1B; Dunn's test, adjusted $P < 0.05$). Compared

with participants from Uganda, vaccinated participants from the United States were more likely to be responsive to ADCP Clade B and C antigens (Fig. 1B; Fisher's exact test, adjusted $P < 0.05$). Participants from the United States were also more likely to respond to ADCP clades A, B, and C compared to participants from Rwanda (Fig. 1B; Fisher's exact test, adjusted $P < 0.05$). Vaccinated participants from Rwanda and Uganda exhibited similar ADCP responsiveness for all clades (Fig. 1B; Fig. S1B).

Cellular immune responses were also evaluated by interferon-γ ELISpot assays against Env PTE peptide pools and against vaccine-matched peptide pools. Participants from the United States consistently exhibited a broader T-cell response (Fig. 1C) and more spot-forming units (SFUs) per million peripheral blood mononuclear cells to nearly all tested peptides than participants from Uganda or Rwanda (Fig. S1C; Dunn's test, adjusted $P < 0.05$). Vaccinated participants from the United States were more likely to exhibit a response to Env Mos2, Gag Mos1, Gag Mos2, Pol Mos1, Pol Mos2, Env1 PTE, Gag PTE, and Pol PTE than participants from both East African countries (Fig. 1C; Fisher's exact test, adjusted $P < 0.05$). Mos1 includes Clade B-like mosaic antigens, Mos2 includes Clade C-like mosaic antigens, and PTE pools are a cross-clade mix of globally relevant peptides representing overall HIV diversity. These data indicate that participants from the United States exhibit broadly enhanced B and T cell responses to HIV-1 vaccination compared to participants from Uganda and Rwanda. Based on these observed differences, we investigated the interactions between the enteric microbiota and the Ad26-based HIV-1 vaccine.

## Ad26-based HIV-1 vaccination is not associated with alterations to the enteric bacterial or bacteriophage microbiome

We first sought to determine whether the administration of the HIV-1 vaccine regimens in the APPROACH study could alter the endogenous enteric bacteriome. We exclusively assessed the US and Ugandan cohorts that had been longitudinally sampled. Representatives from six bacterial phyla (Actinomycetota, Bacteroidota, Bacillota, Pseudomonadota, Verrucomicrobiota, and Fusobacteriota) were identified in both US and Ugandan samples, in addition to species from Elusimicrobiota and Spirochaetota, which were specific to the Ugandan cohort (Fig. 2A and B). Comparing early and late post-vaccination samples revealed concordance in both alpha and beta diversity between time points in both the US and Ugandan cohorts (Fig. 2C through E, paired Wilcoxon signed-rank test and permutational multivariate analysis of variance [PERMANOVA], all $P > 0.05$). There were also no significant changes in bacteriophage composition between early and late time points in either the US and Ugandan cohorts (Fig. S2A through D, paired Wilcoxon signed-rank test and PERMANOVA, all $P > 0.05$).

While there were only limited placebo samples in the US ($n = 4$) and Rwandan cohorts ($n = 6$), we evaluated whether receipt of any HIV-1 vaccine regimen was associated with enteric bacteria (Fig. S2E through H, paired Wilcoxon signed-rank test and PERMANOVA, all $P > 0.05$) or bacteriophage (Fig. S2I through L, paired Wilcoxon signed-rank test and PERMANOVA, all $P > 0.05$) compositions distinct from placebo groups. Overall, these results support that Ad26-based HIV-1 vaccination is not associated with significant alterations to the bacterial or bacteriophage microbiota either across groups or within individuals.

## Geography is a driver of differences in both early and late post-vaccination microbiota samples

Numerous prior reports have documented the important role of geography as a driver of enteric microbiota differences (60–63). Therefore, we analyzed whether the geographic sources of our samples contributed to substantial differences in bacterial and bacteriophage composition. We first compared the bacterial compositions of late post-vaccination samples as this was the only time point with samples from all three regions. Late post-vaccination samples from Uganda and the United States had similar phylum-level compositions (Fig. 3A). However, samples from Rwanda were unique in containing taxa

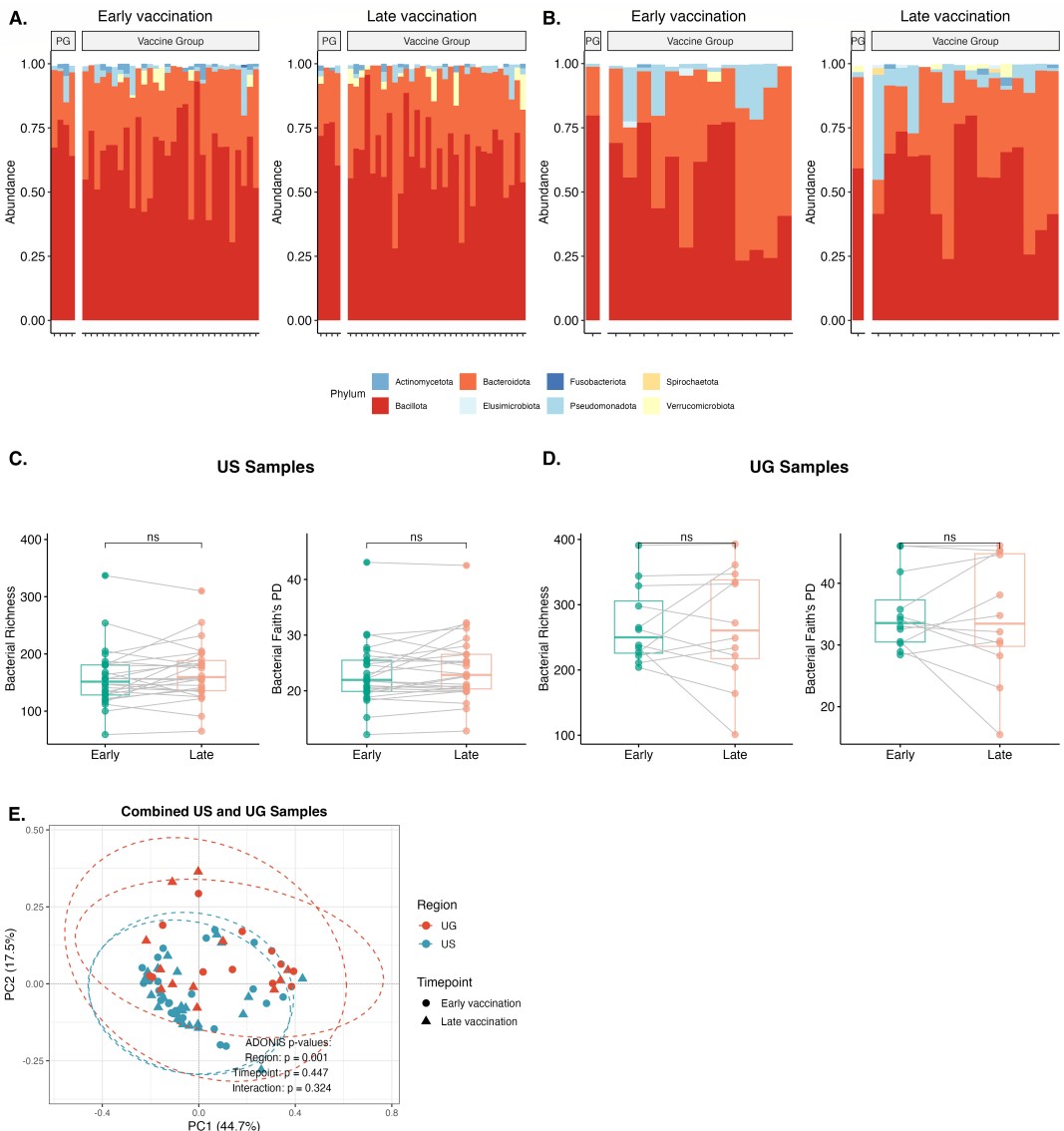

**FIG 2** APPROACH vaccine regimens do not alter the enteric bacterial microbiome. 16S rRNA gene amplicon surveys were performed on fecal samples from participants in the APPROACH vaccine study. Relative abundance of bacterial taxa at the phylum level in placebo (PG) and vaccine groups from (A) the United States (US) and (B) Uganda (UG) at early (week 2) and late (week 26) time points in the vaccination regimen. Each vertical bar corresponds to a study participant. Bacterial richness and Faith's PD were pair-matched and compared between early and late post-vaccination samples (placebo group excluded) from (C) the US and (D) UG. Each point corresponds to a single individual in the vaccine group. Median values and interquartile ranges are shown. Principal coordinates analysis using weighted UniFrac distances was performed by comparing samples from (E) the US UG sample types, which are colored by region, with time points denoted by a symbol. Statistical significance for paired samples was determined using the paired Wilcoxon signed-rank test. Differences between groups in principal coordinates analysis (PCoA) plots were assessed using PERMANOVA. ns, not significant.

from the phylum Spirochaetota. Samples from the United States exhibited significantly lower bacterial richness compared to both Rwandan and Ugandan samples, with richness being comparable between both African cohorts (Fig. 3B, richness: Dunn's test, Rwanda vs Uganda, $P = 0.542$; Rwanda vs United States, $P = 4.96 \times 10^{-10}$; and Uganda vs United States, $P = 7.28 \times 10^{-5}$). Similarly, bacterial diversity (as measured by Faith's PD) was also lower in the US samples when compared to both the African cohorts, which had comparable bacterial diversity (Fig. 3B, Faith's PD: Dunn's test, Rwanda vs Uganda, $P = 0.420$; Rwanda vs United States, $P = 2.55 \times 10^{-10}$; and Uganda vs United States, $P = 8.86 \times 10^{-5}$). Analysis of beta diversity also indicated that all three cohorts contained distinct bacterial community compositions (Fig. 3C, PERMANOVA: Rwanda vs Uganda, $P = 0.003$;

Rwanda vs United States, $P = 0.001$; and Uganda vs United States, $P = 0.001$). Similarly, bacteriophage composition was regionally distinct (Fig. 3D, PERMANOVA: Rwanda vs Uganda, $P = 0.012$; Rwanda vs United States, $P = 0.001$; and Uganda vs United States, $P = 0.001$). Differential abundance testing of 16S rRNA ASVs using DESeq2 revealed that ASVs from the phylum Bacteroidota were more frequently associated with samples from both East African cohorts than from the United States (Fig. 3E and G; Table S4). There were 1.5 times as many Bacteroidota ASVs associated with samples from Rwanda than the United States (Rwanda = 158 and United States = 107) (Fig. 3F), but equivalent numbers of Bacteroidota ASVs associated with Uganda and the United States (Uganda = 138 and United States = 136) (Fig. 3H). There were significantly more ASVs from the phylum *Bacillota* associated with samples from Rwanda than the United States (Rwanda = 682 and United States = 505) (Fig. 3F), but more equivalent numbers of differentially associated *Bacillota* ASVs between the United States and Uganda (Uganda = 562 and United States = 607) (Fig. 3H). More ASVs from the phylum Pseudomonadota were associated with both African cohorts when compared to the United States (Rwanda vs United States = 2.07:1, Uganda vs United States = 20:19) (Fig. 3F and H). When comparing the two East African cohorts, minor differences identified were the frequency of differential ASVs from Bacteroidota (Rwanda = 1.16 and Uganda = 1) and Bacillota (Rwanda = 1.21 and Uganda = 1) phyla (Fig. 3J).

To characterize the microbial differences between geographical regions, we compared DESeq2 and LEfSe analytical approaches for identifying differentially abundant bacterial operational taxonomic units (OTUs) (Table S5). We found substantial variation in the taxa identified by each method across regional comparisons. In the Rwanda-Uganda comparison, DESeq2 exclusively identified 4 OTUs, while LEfSe exclusively identified 447, with 62 OTUs detected by both methods. The Rwanda-US comparison yielded 63 OTUs unique to DESeq2, 143 unique to LEfSe, and 366 OTUs identified by both methods. Similarly, in the US-Uganda comparison, 9 OTUs were exclusively identified by DESeq2, 219 by LEfSe, and 99 by both methods. Given the consistently higher number of OTUs identified solely by LEfSe across all comparisons and concerns about potential false positives, we selected DESeq2 for subsequent analyses. DESeq2 employs a more conservative statistical approach using the negative binomial distribution to model count data, providing robust detection of differentially abundant taxa ($P < 0.05$ by Wald test), while reducing the likelihood of type I errors when comparing microbial communities across these geographical regions. Therefore, we chose to use DESeq2 for our additional analyses.

Analysis of bacterial communities between the United States and Uganda at early time points revealed broadly similar trends (Fig. S3). While common enteric bacterial phyla were present in samples from both countries (Fig. S3A), bacterial richness was significantly lower in samples from the United States (Fig. S3B, Wilcoxon rank-sum test, $P = 1.24 \times 10^{-5}$). Faith's PD was also lower in US samples (Fig. S3B, Wilcoxon rank-sum test, $P = 2.08 \times 10^{-7}$), while the bacteria in US samples are more evenly distributed than in UG samples (Pielou's evenness) (Fig. S3B, Wilcoxon rank-sum test, $P = 0.0262$). Furthermore, early post-vaccination samples from Uganda and the United States exhibited distinct bacterial (Fig. S3C, PERMANOVA, $P = 0.006$) and bacteriophage (Fig. S3D, PERMANOVA, $P = 0.001$) compositions. Samples from Uganda contained a greater representation of differentially abundant Bacteroidota ASVs than samples from the US participants, while samples from the United States exhibited a greater number of differential Bacillota ASVs (Fig. S3E and F). Taken together, these data suggest that the composition of enteric bacterial communities is unique to each region, with more similarity between samples from the two East African cohorts (Rwanda and Uganda) compared to the less diverse samples from the US cohort.

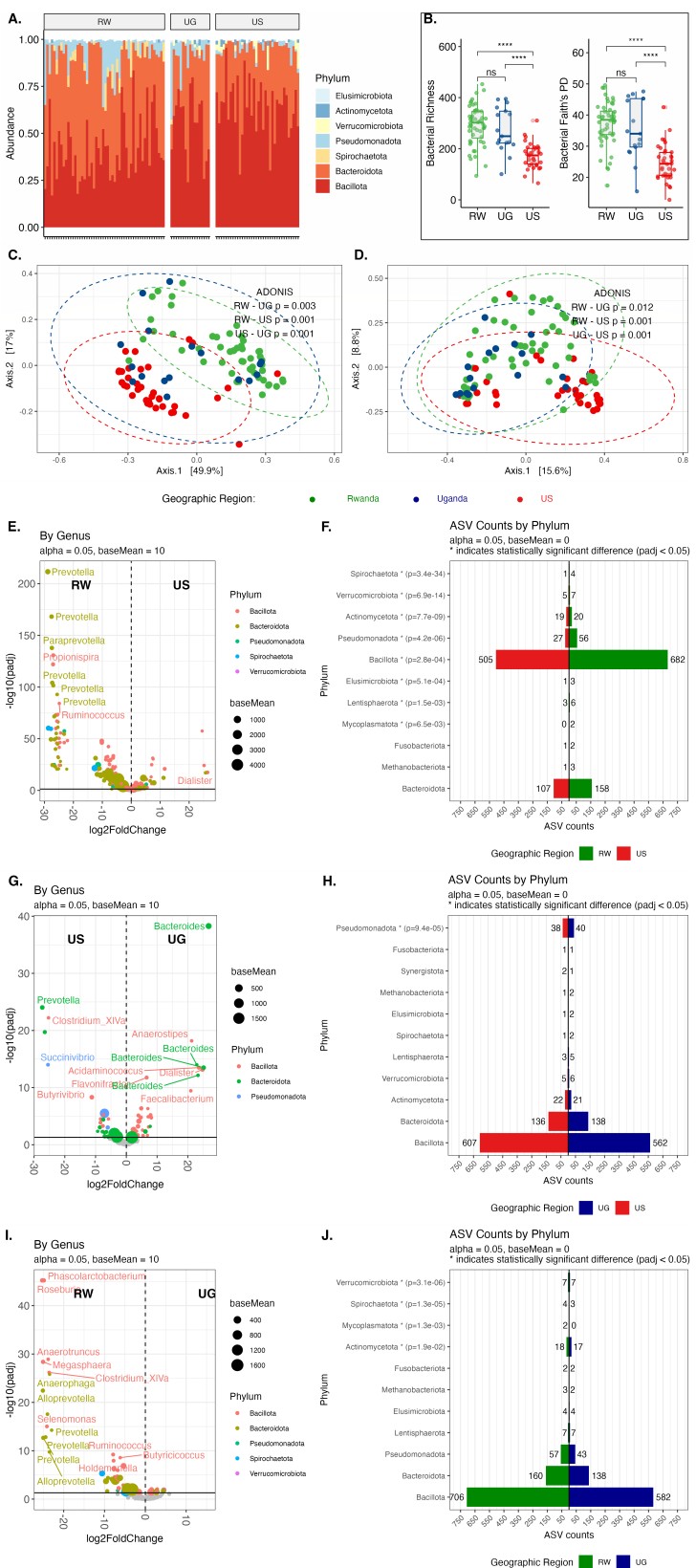

**FIG 3** Geography is a major driver of differences in enteric bacterial communities. 16S rRNA gene amplicon surveys were performed on fecal samples collected from individuals from the United States (US), Uganda (UG), and Rwanda (RW) at the late post-vaccination time point (week 26). (A) Relative

Fig 3 (Continued)

abundance of bacterial taxa at the phylum level for samples from individuals from all three regions. Each vertical bar corresponds to a study participant. (B) Bacterial richness and Faith's PD comparisons between regions. (C) Principal coordinates analysis (PCoA) using weighted UniFrac distances comparing regions. (D) Dimensional reduction (NMDS) of the Bray-Curtis distance for bacteriophage comparing regions. DESeq2 analysis and phylum-level summary counts of taxa differentially abundant between (E and F) RW vs US, (G and H) UG vs US, and (I and J) RW vs UG. Statistical significance in alpha diversity between regions was determined using Dunn's test and adjusted with the Benjamini-Hochberg procedure. Differences between groups in PCoA plots were assessed using PERMANOVA. ****$P < 0.0001$; ***$P < 0.001$; **$P < 0.01$; *$P < 0.05$; and ns, not significant.

## Associations between bacterial communities and HIV-1 clade- and antigen-specific immune responses

Considering the region-specific nature of both HIV-1 Ad26-based vaccine immune responses (Fig. 1) and bacterial communities (Fig. 3), we sought to characterize how enteric bacterial community composition and specific bacterial taxa correlate with regional differences to HIV-1 clade-specific antibodies, antibody functionality, and HIV-1-specific cellular immune responses. We assessed the association between bacterial alpha diversity (both richness and Faith's PD) and HIV-1 vaccine immune responses with all regions combined (general) or between regions (regional) (Fig. 4; Fig. S4). There was a general negative correlation between bacterial alpha diversity and both ADCP phagocytic scores and ELISpot SFUs except for the Env2/3 PTE antigens (Fig. 4A), with many of these correlations reaching statistical significance. There were no significant general correlations between bacterial alpha diversity measurements and ELISA titers for any tested antigen (Fig. 4A). While there were several positive and negative correlation coefficients between bacterial alpha diversity and immune responses within each region, they were all non-significant (Fig. 4A). Comparison of linear regressions between each region using analysis of covariance (ANCOVA) revealed several regional associations between bacterial alpha diversity measures and various immune responses (Fig. 4B through D; Fig. S4A through C). All significant associations were between the lower diversity samples from the United States and increased immune responsiveness, including increased ELISA titers to clade B and clade C (Con C) antigens, increased ADCP phagocytic scores to Mos1 and clade C (Con C), and increased ELISpot SFU to Env and Gag Mos1, Pol Mos1/2, Gag PTE, and Pol PTE, compared to either one or both East African regions. These findings suggest that the general correlation observed between bacterial alpha diversity and specific immune responsiveness is driven by regional differences in diversity and dominated by the low diversity samples from the United States.

## Associations between specific bacterial taxa and HIV-1 clade- and antigen-specific immune responses

Next, we sought to identify bacterial ASVs associated with HIV clade- and antigen-specific immune responses within each region (Fig. 5). Due to the larger number of available samples per region, we focused on comparing immune assay responders to non-responders for Rwanda and the United States at the late post-vaccination time point. We identified numerous differentially abundant ASVs associated with response or non-response to individual immune assays for both cohorts (Fig. S5A and B). We grouped these ASVs by phylum and assessed whether, based on the proportion of phylum-level ASVs present in the samples, a disproportionate number of differentially abundant ASVs were associated with response or non-response by Fisher's exact test. Specifically, we found that several bacteria from the phylum Bacteroidota were associated with diminished immune responsiveness for Rwandan samples to the dominant circulating strain of that region, clade A (two *Spongiimonas flava*, *Prevotellamassilia timonensis*, four *Prevotella copri*, *Prevotella buccalis*, *Prevotella* sp., *Phocaeicola dorei*, two *Pedobacter* sp., *Marinilabilia* sp., *Barnesiella intestinihominis*, *Bacteroides fragilis*, *Bacteroides clarus*,

**A.**

| Immune assay | | Bacterial richness multiple correlation test | | | | | |
|---|---|---|---|---|---|---|---|
| | | Total sample | | | RW | US | UG |
| | Clade | Correlation coefficients | Adjusted p-value | Adjusted p-signif | Correlation coefficients | Correlation coefficients | Correlation coefficients |
| ELISA | Clade A | -0.19 | 0.17 | ns | 0.13 | -0.25 | 0.15 |
| | Clade B | -0.22 | 0.17 | ns | 0.04 | -0.19 | -0.05 |
| | Clade C | -0.16 | 0.17 | ns | 0.05 | -0.03 | 0.13 |
| | Clade C (Con C) | -0.21 | 0.17 | ns | 0.04 | -0.29 | 0.04 |
| | Mos1 | -0.22 | 0.17 | ns | -0.25 | -0.14 | 0.09 |
| ADCP | Clade A | -0.26 | 0.01 | * | -0.23 | -0.23 | 0.24 |
| | Clade B | -0.28 | 0.01 | * | -0.21 | -0.3 | 0.4 |
| | Clade C | -0.37 | 0.001 | ** | -0.31 | -0.27 | 0.4 |
| | Clade C (Con C) | -0.3 | 0.009 | ** | -0.07 | -0.3 | 0.52 |
| | Mos1 | -0.36 | 0.001 | ** | -0.34 | -0.28 | 0.25 |
| ELISpot | Env Mos1 | -0.41 | 4.34e-04 | *** | -0.19 | -0.14 | -0.25 |
| | Env Mos2 | -0.31 | 0.01 | * | -0.1 | -0.29 | -0.04 |
| | Gag Mos1 | -0.3 | 0.02 | * | -0.01 | -0.12 | -0.2 |
| | Gag Mos2 | -0.26 | 0.03 | * | -0.1 | -0.08 | -0.2 |
| | Pol Mos1 | -0.47 | 2.00E-05 | **** | -0.12 | -0.12 | -0.05 |
| | Pol Mos2 | -0.51 | 2.00E-06 | **** | -0.31 | -0.09 | -0.12 |
| | Env1 PTE | -0.33 | 0.007 | ** | -0.26 | -0.12 | -0.18 |
| | Env2 PTE | -0.23 | 0.05 | ns | -0.18 | -0.03 | 0.01 |
| | Env3 PTE | -0.22 | 0.05 | ns | -0.07 | 0.002 | 0.08 |
| | Gag PTE | -0.34 | 0.006 | ** | -0.14 | -0.06 | -0.31 |
| | Pol PTE | -0.44 | 8.11E-05 | **** | -0.29 | -0.1 | -0.11 |

| Immune assay | | Bacterial Faith's PD multiple correlation test | | | | | |
|---|---|---|---|---|---|---|---|
| | | Total sample | | | RW | US | UG |
| | Clade | Correlation coefficients | Adjusted p-value | Adjusted p-signif | Correlation coefficients | Correlation coefficients | Correlation coefficients |
| ELISA | Clade A | -0.24 | 0.11 | ns | 0.06 | -0.25 | 0.1 |
| | Clade B | -0.23 | 0.11 | ns | 0.02 | -0.19 | -0.02 |
| | Clade C | -0.23 | 0.11 | ns | 0.001 | -0.03 | 0.21 |
| | Clade C (Con C) | -0.23 | 0.11 | ns | 0.04 | -0.29 | 0.08 |
| | Mos1 | -0.18 | 0.11 | ns | -0.27 | -0.18 | 0.13 |
| ADCP | Clade A | -0.27 | 0.01 | * | -0.18 | -0.28 | 0.16 |
| | Clade B | -0.28 | 0.01 | * | -0.15 | -0.36 | 0.35 |
| | Clade C | -0.39 | 0.0006 | *** | -0.26 | -0.32 | 0.42 |
| | Clade C (Con C) | -0.33 | 0.003 | ** | -0.05 | -0.34 | 0.45 |
| | Mos1 | -0.36 | 0.001 | ** | -0.3 | -0.36 | 0.24 |
| ELISpot | Env Mos1 | -0.41 | 4.17E-04 | *** | -0.19 | -0.12 | -0.14 |
| | Env Mos2 | -0.31 | 0.01 | * | -0.11 | -0.19 | 0.01 |
| | Gag Mos1 | -0.34 | 0.006 | ** | -0.06 | -0.2 | -0.11 |
| | Gag Mos2 | -0.3 | 0.01 | * | -0.18 | -0.15 | -0.08 |
| | Pol Mos1 | -0.49 | 5.67E-06 | **** | -0.13 | -0.14 | 0.09 |
| | Pol Mos2 | -0.53 | 5.82E-07 | **** | -0.37 | -0.1 | 0.02 |
| | Env1 PTE | -0.29 | 0.01 | * | -0.26 | 0.02 | -0.07 |
| | Env2 PTE | -0.22 | 0.06 | ns | -0.14 | -0.02 | 0.01 |
| | Env3 PTE | -0.22 | 0.06 | ns | -0.08 | 0.01 | 0.12 |
| | Gag PTE | -0.36 | 0.003 | ** | -0.15 | -0.1 | -0.19 |
| | Pal PTE | -0.46 | 2.84E-05 | **** | -0.29 | -0.14 | 0.05 |

**B.**

**C.**

**D.**

FIG 4 Association between bacterial alpha diversity and HIV-1 Ad26-based vaccine immune responses. 16S rRNA gene amplicon surveys were performed on fecal samples collected from individuals from the United States (US), Uganda (UG), and Rwanda (RW) at the late post-vaccination time point (week 26). Bacterial alpha diversity (richness and Faith's PD) was calculated per sample and compared to clade- and antigen-specific immune assays using both Spearman's

Fig 4 (Continued)

rank-order correlation analysis and linear model comparisons using ANCOVA. (A) Summary of multiple correlations of bacterial richness or Faith's PD and HIV-1 Ad26-based vaccine immune responses. Both all regions combined (total sample) and individual regions (RW, US, and UG) were analyzed. ANCOVA comparing bacterial richness and Faith's PD to (B) ELISA titers, (C) ADCP scores, and (D) ELISpot responses. Only assays with significance ($P < 0.05$) between one or more regions are shown. Color lines depict the linear model with gray areas indicating the standard error of the mean for each group. The dotted line indicates the lower limit of quantification threshold (B), the limit of detection threshold (C), and the 95th percentile of the overall baseline values (D) for the indicated assays. ****$P < 0.0001$; ***$P < 0.001$; **$P < 0.01$; *$P < 0.05$; and ns, not significant. PTE, potential T-cell epitope; PBMC, peripheral blood mononuclear cells; and SFU, spot-forming units.

and *Bacteroides caccae*) (Fig. 5A; Fig. S5A). Two Bacteroidota species were significantly associated with increased responsiveness to the clade A vaccine in Rwanda (*Spongiimonas flava* and four *Prevotella copri*) (Fig. 5A; Fig. S5A). In addition, Bacteroidota ASVs were associated with diminished antigen-specific immune responses to Gag Mos2 (*Prevotellamassilia timonensis*, two *Prevotella stercorea*, two *Prevotella copri*, *Prevotella brevis*, and *Bacteroides clarus*) as well as Gag PTE peptide pools in Rwandan samples (*Prevotellamassilia timonensis*, two *Prevotella stercorea*, two *Prevotella copri*, *Prevotella brevis*, and *Bacteroides clarus*), but increased responses to Env Mos1 (*Spongiimonas flava*, three *Prevotellamassilia timonensis*, *Prevotella copri*, *Pedobacter* sp., and *Massiliprevotella massiliensis*) (Fig. 5A; Fig. S5B). Some Bacteroidota ASVs were associated with diminished antigen-specific immune responses to Env2 PTE peptide pools (*Bacteroides fragilis* and *Bacteroides caccae*), while others were associated with increased responses (*Prevotellamassilia timonensis* and *Prevotella copri*) (Fig. 5A; Fig. S5B). In contrast, Bacteroidota ASVs were not associated with non-responsiveness to any clade in US samples and instead were associated with increased ADCP responsiveness to Mos1 (*Roseburia* sp., *Phocaeicola massiliensis*, *Phocaeicola dorei*, *Parabacteroides merdae*, *Flintibacter butyricus*, *Bacteroides uniformis*, and *Bacteroides* sp.) (Fig. 5A; Fig. S5A). Increased US sample immune responsiveness was more frequently associated with ASVs from the phylum Bacillota, including increased ADCP responses to Mos1 (*Subdoligranulum variabile*, *Sprobacter termitidis*, *Ruminococcus bromii*, *Ruminiclostridium cellobioparum*, *Roseburia faecis*, *Oscillibacter valericigenes*, *Negativibacillus massiliensis*, *Hungatella xylanolytica*, *Flintibacter butyricus*, *Faecalibacterium prausnitzii*, *Eubacterium coprostanoligenes*, *Dorea longicatena*, *Dialister invisus*, and *Coprococcus eutactus*) as well as cellular immune responses and Pol Mos1 (*Subdoligranulum variabile*, *Sprobacter termitidis*, *Ruminococcus bromii*, *Ruminiclostridium cellobioparum*, *Roseburia faecis*, *Oscillibacter valericigenes*, *Negativibacillus massiliensis*, *Hungatella xylanolytica*, *Flintibacter butyricus*, *Faecalibacterium prausnitzii*, *Eubacterium coprostanoligenes*, and *Coprococcus eutactus*). Env Mos2 was associated with both increased (*Phascolarctobacterium succinatutens*, *Holdemanella biformis*, *Faecalibacterium prausnitzii*, *Dialister propionicifaciens*, *Coprococcus eutactus*, and *Allisonella histaminiformans*) and decreased cellular immune responses to Env Mos2 (*Ruminococcus bromii*, *Peptococcus simiae*, *Monoglobus pectinilyticus*, *Megasphaera hexanoica*, *Megamonas funiformis*, *Lactobacillus rogosae*, *Lachnospira pectinoschiza*, *Lacnospira eligens*, *Holdemanella biformis*, five *Faecalibacterium prausnitzii*, *Dialister invisus*, *Ruminococcus* sp., and two *Eubacterium rectale*) (Fig. 5B; Fig. S5B). These ASVs are captured in Table S6. Overall, these findings suggest distinct region-specific associations between both phyla and numerous specific ASVs with enhanced or diminished responsiveness to specific HIV-1 antigens and epitopes.

## Enteric viruses are more common in East African samples than in the United States and are associated with immune responsiveness to clade-specific immune responses in Rwanda

As vertebrate virus co-infection has been associated with immunological outcomes to other vaccines (38–40), we assessed both the presence and abundance of viruses known to infect vertebrates in US and East African samples. Analysis of the RNA and DNA vertebrate virome of samples from all three regions identified contigs from six viral

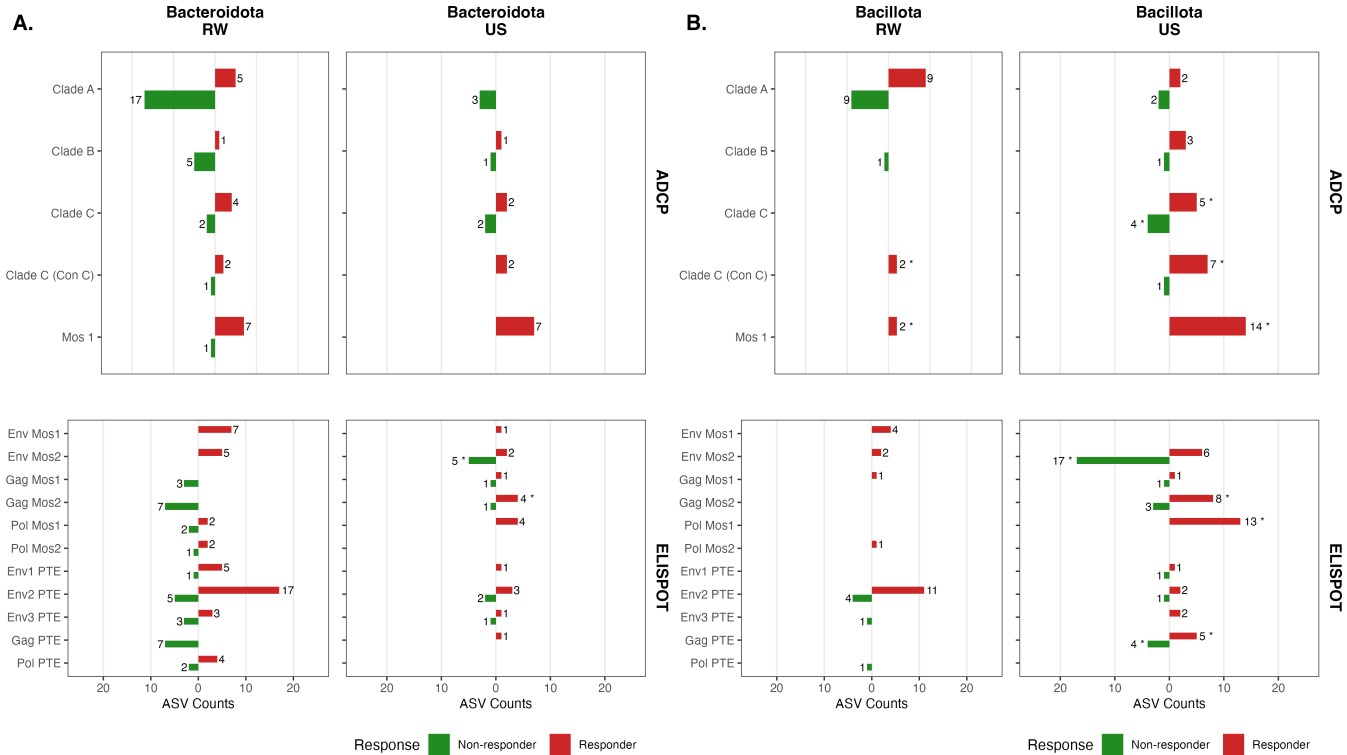

**FIG 5** *Bacteroidota* and *Bacillota* ASVs are associated with altered HIV-1 clade and antigen-specific immune responses. Differential abundance testing was used to identify ASV associated with clade- and peptide pool-specific immune responses using ADCP and ELISpot assays using the late post-vaccination time point (week 26) samples from Rwanda (RW) and Uganda (UG). Vaccine response was defined as a value more than the threshold (if baseline is less than the threshold or is missing); otherwise, it was defined as a value with a three-time increase from baseline (if baseline is greater than or equal to the threshold) for Env-specific gp140 antibody-dependent phagocytic cells or antibody titers (ADCP). Vaccine response was defined as geometric mean titers of greater than 200 spot-forming cells to one or more HIV antigens (ELISpot) as described previously (21, 22, 42–44). (A) Summary of differentially abundant *Bacteroidota* and (B) *Bacillota* ASVs per immunoassay. Significance was determined using Fisher's exact test and adjusted for multiple comparisons using the Benjamini-Hochberg procedure. ****$P < 0.0001$; ***$P < 0.001$; **$P < 0.01$; and *$P < 0.05$.

families: *Adenoviridae, Caliciviridae, Astroviridae, Parvoviridae, Circoviridae,* and *Picornaviridae* (Fig. 6A). Only two vertebrate virus families were detected in samples from the United States (*Adenoviridae* and *Parvoviridae*), while all six families were detected in samples from both Rwanda and Uganda. Four of these families contained only a single representative viral genus: *Adenoviridae* contained only *Mastadenovirus, Circoviridae* contained only *Circovirus, Caliciviridae* contained only *Norovirus,* and *Astroviridae* contained only *Mamastrovirus.* Two genera of *Parvoviridae, Scindoambidensovirus* and *Ambidensovirus,* were detected in samples from East Africa. Both of these are likely diet-associated as they are most frequently detected in insects, many of which feed on crops and crustaceans (64). We also detected reads from two genera of *Picornaviridae, Cosavirus* and *Enterovirus.* We compared the abundances of each viral family between each region (Fig. 6B). Both *Picornaviridae* and *Circoviridae* were more abundant in Rwanda and Uganda when compared to the United States, while *Parvoviridae* and *Astroviridae* were more abundant in Uganda than in the United States and Rwanda. Comparison of matched early and late post-vaccination samples for all viral families indicated that vaccination did not alter viral abundance (Fig. S6A through D).

Finally, we compared the abundance of each viral contig against all measured HIV-1 clade and antigen-specific immune measures. Two significant associations were identified in samples from Rwanda (Fig. S6E and F). The abundance of *Cosavirus* was significantly associated with responsiveness to ADCP clade C (Con C) in Rwandan samples, and *Enterovirus* abundance was significantly associated with ADCP clade A responsiveness in Rwandan samples (Wilcoxon signed-rank test $P < 0.05$). Further

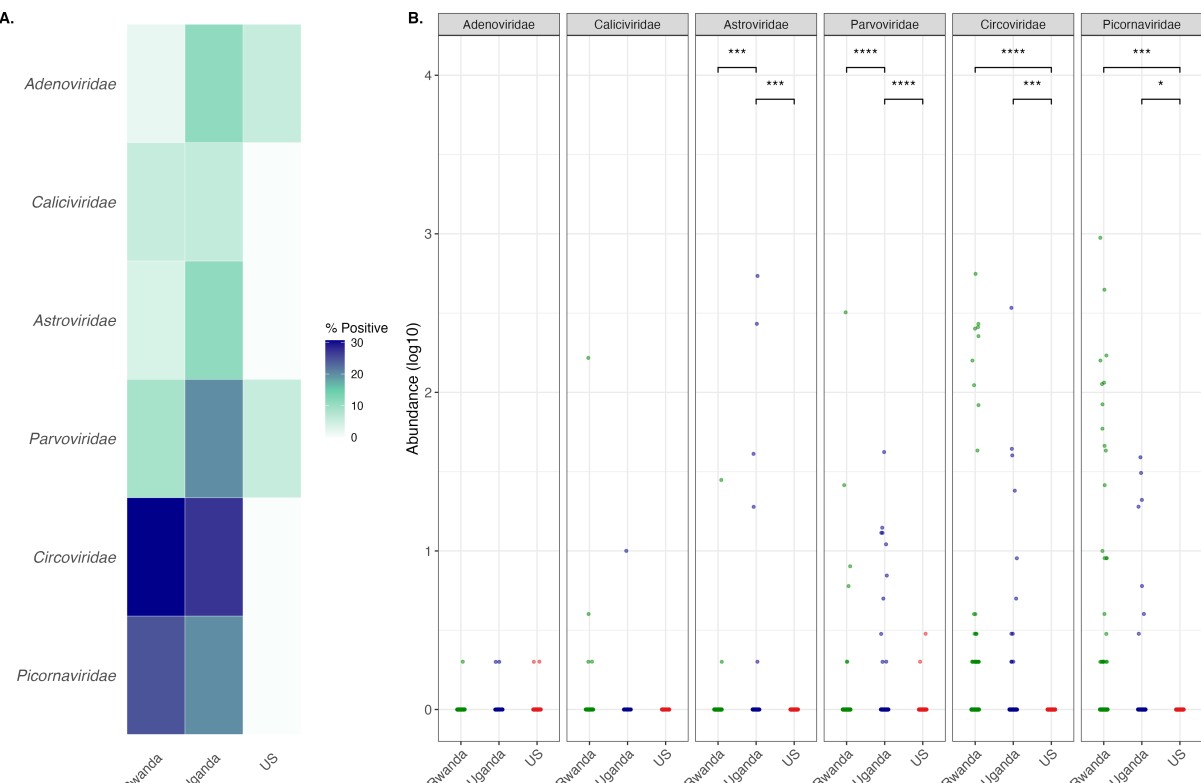

**FIG 6** Vertebrate viruses are more diverse in fecal samples from East Africa than in samples from the United States. Vertebrate virus analysis was performed on fecal samples collected from individuals from the United States, Uganda, and Rwanda at the late post-vaccination time point (week 26). (A) Vertebrate viral families identified across all regions. (B) Comparative abundance of each viral family in samples from each region. Significance was determined using Dunn's test and adjusted for multiple comparisons using the Holm method. ****$P < 0.0001$; ***$P < 0.001$; **$P < 0.01$; *$P < 0.05$; and ns, not significant.

analysis of the *Cosavirus* contigs indicates that it is most closely related to other unclassified *Cosavirus* species in reference databases and that the enteroviral contig is most similar to a non-polio enterovirus C, human enterovirus 113. Overall, these data indicate enhanced vertebrate viral diversity present in samples from East African nations compared to the United States, consistent with a broad negative association of bacterial and viral diversity with immune responsiveness to vaccination.

## DISCUSSION

To gain insight into the interplay between gut microbial populations and the adaptive immune response to systemically administered vaccines, we characterized the fecal bacteriome and vertebrate virome in healthy individuals from Uganda, Rwanda, and the United States who were administered mosaic Ad26-based HIV-1 vaccine regimens. These mosaic Ad26-based vaccine regimens have been demonstrated to be safe and well-tolerated and to induce robust humoral and cellular immune responses in healthy participants from different countries (21). Importantly, there are substantial differences in the immunological responses between participants from different geographic regions in terms of both antibody functionality and HIV-1-specific cellular immune responses. There is more limited vaccine immunogenicity in individuals from East African countries compared to those living in the United States, and significantly higher cross-clade-binding antibodies are elicited in individuals from Rwanda compared to Uganda.

Chronic inflammatory conditions, such as inflammatory bowel diseases or metabolic syndromes, are associated with dysbiosis of the microbiota (54, 65–67). Systemic inflammation and intestinal mucosal damage, observed with progressive SIV and HIV infection, are also associated with changes in the enteric virome and bacterial

microbiome (41, 42, 68). Contrary to these chronic inflammatory stimuli, and consistent with our expectations for a systemically administered vaccine, we did not find any modulation of the bacteriome or prokaryotic and vertebrate viromes associated with vaccination. These findings are consistent with our prior observation that administration of an Ad26-based vaccination against SIV largely abrogates the enteric virome expansion associated with SIV infection (43), indicating that vaccination can protect against disease-associated microbiota changes without independently altering microbial communities in healthy individuals. Our data further support the finding that Ad26-based HIV-1 vaccination does not affect the intestinal microbiota regardless of the geographic location of those vaccinated, supporting the safety of broad distribution.

Geography has been previously identified as a major driver of microbiota differences, with significant variation observed between samples obtained from individuals from different countries (60, 69, 70). Recent studies suggest that differences in the intestinal microbiota between geographically distinct cohorts may be driven predominantly by diet, lifestyle habits, and environment as opposed to host genetics, as the microbiomes of new US immigrants from LMICs undergo rapid changes and become more similar to Westernized populations, with a loss of diversity, replacement of *Prevotella* species with *Bacteroidaceae*, and a loss of bacterial enzymes associated with plant fiber degradation (71, 72). Consistent with these prior reports, we observed that the bacteriome and bacteriophage microbiome were markedly different between individuals from the United States, Uganda, and Rwanda. Bacterial alpha diversity was significantly higher in samples from both Uganda and Rwanda cohorts compared to the United States. Despite broad similarities in phylum-level composition of bacterial communities, beta diversity analyses revealed dramatic compositional differences between samples from different countries at both early and late post-vaccination time points. In addition, the fecal microbiota composition of individuals from Rwanda exhibiting pronounced enrichment of Spirochaetota and higher Bacteroidota:Bacillota ratios was consistent with prior reports of microbiota characteristics of individuals from traditional rural societies (69, 70, 73). We also observed numerous vertebrate virus families exclusively present in samples from Rwanda and Uganda, indicating enhanced diversity of vertebrate viruses in adults from East African nations compared to the United States. Thus, geography was a major determinant of microbiota variation in our study, emphasizing the importance of analyzing samples to assess the interactions of vaccination and the microbiota separately, depending on their geographical source.

After decades-long efforts to develop an efficacious preventative HIV-1 vaccine, the safety and immunogenicity of several candidate HIV-1 vaccination regimens have been globally evaluated in phase I/II clinical trials (6). However, many vaccines are less immunogenic than expected when tested in LMICs, a variation that may be influenced by many factors, including environmental factors (geographic location, season, family size, and toxins), behavioral factors (smoking, alcohol consumption, exercise, and sleep), nutritional factors (diet, body mass index, micronutrients, and enteropathy), perinatal factors (gestational age, birth weight, feeding method, and maternal factors), intrinsic host factors (age, sex, and genetics), and extrinsic factors (infections and antibiotics) beyond the microbiota (74–76). Furthermore, vaccine factors (vaccine type, adjuvant, and dose) and administration factors (schedule, site, and route) also play important roles in the individual's' response to vaccines. Although many strategies have been developed to improve vaccine efficacy and application throughout the world, our understanding of how the intestinal microbiota may affect vaccine immunogenicity and efficacy remains elusive (75, 77). In this study, we found, consistent with prior reports, more limited vaccine immune responsiveness in participants from LMICs compared to HICs (24–26), decreased breadth and strength of antibody titers, antibody functionality, and cellular immune responses in participants from Uganda and Rwanda compared to the United States, as measured by ELISA, ADCP, and IFN-γ ELISpot assays. More recently, the mosaic Ad26-based HIV vaccine was tested in young women in Sub-Saharan Africa. This trial, called the "Imbokodo" clinical trial, showed that the investigational HIV vaccine regimen

did not provide sufficient protection against HIV infection in a population of young women in sub-Saharan Africa at high risk of acquiring HIV (5). Another more recent phase III trial, the "Mosaico" trial, which evaluated this vaccine in cisgender men and transgender people who have sex with cisgender men and/or transgender people in multiple countries, also indicated that it was not effective at preventing the acquisition of HIV-1 (22). The investigational vaccine was consistently found to have a favorable safety profile with no serious adverse events. However, based on the results presented here, it would be interesting to evaluate if the enteric microbiome in the study population was associated with vaccine efficacy.

The composition and function of the intestinal microbiota are increasingly recognized as crucial, targetable influences on host immune development and overall immune response (23, 28, 78, 79). For instance, studies involving antibiotic-mediated alteration of the bacteriome have demonstrated that gut microbiome composition significantly impacts immunological responses to both oral and parenteral vaccines (30, 31). Here, we were able to directly assess the microbiota of healthy participants in both a HIC and two LMICs in parallel with robust immunophenotyping to assess for microbial associations with vaccine-elicited immunity. We discovered that, across all samples, low bacterial richness and Faith's PD were associated with stronger cross-clade ADCP and T cell responses. These associations were predominantly driven by negative correlations between bacterial alpha diversity and immune responses in the United States, as most samples from Uganda and Rwanda were associated with both high bacterial alpha diversity and low vaccine reactivity and were significantly different from US samples. These findings are contrary to what is often observed in chronic disease states, wherein increased bacterial diversity is typically associated with positive health outcomes (80).

Global HIV-1 genetic diversity is a complex challenge for vaccine development, with dominant HIV-1 subtypes varying with geography (10). In the United States and other HICs, clade B accounts for the majority of infections, while in East African countries, approximately half of infections are caused by clade A, with clade C serving as another major contributor (10). While most US participants exhibited ADCP responses to geographically relevant clade B, ADCP responses to clades A and C were observed in less than half of the East African participants (Fig. 1B). T cell responses to peptide pools followed similar clade-specific patterns (Fig. 1C). Response to Gag and Pol Mos1 (Clade B-like), Mos2 (Clade C-like), and PTE (cross-clade mix of globally relevant peptides representing overall HIV diversity) were more frequent in the US cohort than in either East African nation. Responses to Env were more complex, as the overall response frequency to Env Mos1 was similar across all regions but was more frequent in the United States to Env Mos2. Response to PTE of Env1 and Env3 was more frequent in the United States than in Uganda, but similar to samples from Rwanda. These data suggest not only that immunological response to specific HIV peptides is region-specific (with more frequent responses in the United States than in East Africa) but that they are also peptide-type specific. Strikingly, while rates of response differed to specific peptide pools, the strength of T cell responses to almost all pools, as measured by ELISpot assay, was consistently higher in the United States (Fig. S1C).

In Rwanda, nonresponse to clade A and impaired cellular immune responses to numerous peptide pools were associated with an overrepresentation of specific *Bacteroidota* taxa, while in the United States, functional antibody responsiveness to Mos1 and cellular immune responses were associated with an overrepresentation of Bacillota. The specific taxa are listed in the results of Fig. 5 and Fig. S5. These findings raise important questions about the functional contribution of these phyla, as well as the specific taxa identified within them, to repression or enhancement of immune responses. Further investigation into how these exact species or strains of bacteria may regulate vaccine responsiveness will be an important area for additional future studies.

It has previously been reported that vertebrate viromes vary with geography and socioeconomic status, and that viral pathogens are detected at higher rates in LMICs compared to HICs (81–84). Additionally, based on within-country comparisons,

vertebrate viruses, including *Enterovirus,* have been negatively associated with antibody responses to oral rotavirus and poliovirus vaccines (38–40). Our findings here, wherein the diversity of vertebrate viruses detected was much greater in samples from Rwanda and Uganda compared to the United States, are consistent with these prior findings and support that vertebrate viral diversity is negatively associated with vaccine responsiveness. A recent study in mice, comparing pet shop mice with complex natural pathogen exposure histories to standard pathogen-free mice, shows that natural pathogen exposures are inversely correlated with immune responses to influenza vaccination (32). Similarly, sequential exposure of mice to natural viral pathogens and helminths is associated with decreased antibody responses to yellow fever virus vaccination (85). The associations we report here thus support an emerging paradigm in which exposure to enhanced microbiota and/or pathogen complexity may limit the strength of elicited immune responses to vaccination.

Gut microbes and other environmental antigens may mimic HIV antigens, thereby priming naive B cells toward cross-reactive antigen-specific responses that become immunodominant following HIV vaccination (86, 87). Previous studies have also shown that the populations of memory CD4 T cells with anti-HIV-1 repertoires can be shaped by intestinal microbial antigens and influence the immune response to HIV-1 vaccines (88, 89). Thus, the microbiota may have varying degrees of direct and indirect influence on vaccine responses, especially for HIV-1 vaccines. While some studies have begun to explore the associations between specific bacterial taxa in the enteric microbiota and immune responses to vaccination and/or probiotic administration to enhance these responses (90, 91), much remains to be done to specifically address differences between HICs and LMICs in parenteral HIV-1 vaccine immune responsiveness.

This study has several limitations. Its observational design identifies correlations but does not establish causality, and the proposed mechanisms linking microbiota to immune responses require experimental validation. The pooling of samples across vaccine arms, though necessary for statistical power, represents a limitation as vaccine types were not equally distributed across regions, which could confound the observed microbiome-vaccine response associations. The lack of baseline samples limits our ability to assess pre-vaccination microbiome status and its potential influence on vaccine responses. Not all potential confounding factors influencing vaccine immunogenicity could be comprehensively measured or controlled. Furthermore, fecal microbiome analysis, though informative, may not fully reflect microbial dynamics or immunological interactions occurring at the gut mucosal interface. The findings for this specific Ad26-vectored vaccine may not be generalizable to other vaccine types. Finally, shotgun VLP sequencing faces inherent challenges in detecting all viral taxa, particularly low-abundance RNA viruses, and in accurately assessing their functional activity levels.

## Conclusions

In summary, our study demonstrates that Ad26-based HIV-1 vaccine regimens do not have any modulatory effects on the microbiota of healthy individuals to whom it isthey are administered, regardless of their country of residence. Geography is the major driver of microbiota differences within our cohort, and vaccine-induced antibody levels, antibody functionality, and cellular immune responses are also geographically distinct. Furthermore, our data indicate that differences in both overall bacterial and viral diversity and in specific microbiota taxa between participants from the United States and East African countries correlate with differential immune responses to Ad26-based HIV-1 vaccination. These findings may have important implications for other licensed Ad26-based vaccines, such as those designed to protect against Ebola and SARS-CoV-2 viruses (92, 93). Further investigation into whether specific taxa and/or enhanced microbial richness limit the elicitation of robust vaccine responses, as well as exploration of antibiotic or probiotic approaches to enhance parenteral vaccine immunogenicity in LMICs, will be critical future endeavors.

## ACKNOWLEDGMENTS

We acknowledge the DNA Sequencing Innovation Laboratory of Washington University for assistance in sample sequencing.

This study was supported by NIH grants RC2 DK116713 (S.A.H.), R01 OD024917 (M.T.B., S.A.H., and D.H.B.), R01 AI141716 (M.T.B.), the Children's Discovery Institute of St. Louis Children's Hospital and Washington University (MI-II-2019-790) (M.T.B. and S.A.H.). The funders had no role in the decision to publish or the preparation of the manuscript.

Y.L., M.T.B., and S.A.H. wrote the manuscript. All authors reviewed and edited the manuscript. Bioinformatic analyses were conducted by Y.L., A.H.K., R.R., K.A.M., and L.W. Samples and immune response data were collected and coordinated by D.J.S., M.G.P., O.Y., and D.H.B. Samples were processed and sequenced by L.D. Study design was originally conceived by H.W.V. and D.H.B.

## AUTHOR AFFILIATIONS

[1]Division of Infectious Diseases, Department of Medicine, Washington University School of Medicine, St. Louis, Missouri, USA

[2]Janssen Vaccines and Prevention BV, Leiden, the Netherlands

[3]Department of Pathology and Immunology, Washington University School of Medicine, St. Louis, Missouri, USA

[4]Janssen R&D BE, Antwerp, Belgium

[5]Department of Internal Medicine, UT Southwestern Medical Center, Dallas, Texas, USA

[6]Beth Israel Deaconess Medical Center, Harvard Medical School, Boston, Massachusetts, USA

[7]Ragon Institute of MGH, MIT and Harvard, Cambridge, Massachusetts, USA

[8]Edison Family Center for Genome Sciences & Systems Biology, Washington University School of Medicine, St. Louis, Missouri, USA

## AUTHOR ORCIDs

Yuhao Li http://orcid.org/0000-0003-2413-0937

Andrew HyoungJin Kim http://orcid.org/0000-0003-0971-6024

Kathie A. Mihindukulasuriya http://orcid.org/0000-0001-9372-3758

Herbert W. Virgin http://orcid.org/0000-0001-8580-7628

Dan H. Barouch http://orcid.org/0000-0001-5127-4659

Megan T. Baldridge http://orcid.org/0000-0002-7030-6131

Scott A. Handley http://orcid.org/0000-0002-2143-6570

## FUNDING

| Funder | Grant(s) | Author(s) |
|---|---|---|
| National Institute of Diabetes and Digestive and Kidney Diseases | RC2 DK116713 | Scott A. Handley |
| NIH Office of the Director | R01 OD024917 | Dan H. Barouch |
| | | Megan T. Baldridge |
| | | Scott A. Handley |
| National Institute of Allergy and Infectious Diseases | R01 AI141716 | Megan T. Baldridge |
| Children's Discovery Institute | MI-II-2019-790 | Megan T. Baldridge |
| | | Scott A. Handley |

## AUTHOR CONTRIBUTIONS

Yuhao Li, Formal analysis, Writing – original draft, Writing – review and editing | Daniel J. Stieh, Investigation, Resources, Writing – review and editing | Lindsay Droit, Data curation, Formal analysis, Methodology, Writing – original draft, Writing – review and

editing | Andrew HyoungJin Kim, Formal analysis, Investigation, Resources, Writing – review and editing | Rachel Rodgers, Data curation, Formal analysis, Methodology, Writing – review and editing | Kathie A. Mihindukulasuriya, Formal analysis, Writing – review and editing | Leran Wang, Formal analysis, Writing – review and editing | Maria G. Pau, Methodology, Resources, Writing – review and editing | Olive Yuan, Formal analysis, Methodology, Resources, Writing – review and editing | Herbert W. Virgin, Conceptualization, Formal analysis, Writing – review and editing | Dan H. Barouch, Conceptualization, Funding acquisition, Methodology, Resources, Supervision, Writing – original draft, Writing – review and editing | Megan T. Baldridge, Funding acquisition, Methodology, Resources, Supervision, Writing – original draft, Writing – review and editing | Scott A. Handley, Conceptualization, Funding acquisition, Supervision, Writing – original draft, Writing – review and editing

## DATA AVAILABILITY

The accession number for the data reported in this paper is PRJEB48706. Full analysis workflows for microbiome dada2 ASV resolution, statistical analysis, and plotting are available at: https://github.com/shandley/HIV_VACCINE_STUDY.

## ETHICS APPROVAL

The APPROACH trial (NCT02315703) was approved by institutional review boards at all participating sites.

## ADDITIONAL FILES

The following material is available online.

### Supplemental Material

**Supplemental Figures Part 1 (mSystems01435-25-s0001.pdf).** Figures S1 to S4.
**Supplemental Figures Part 2 (mSystems01435-25-s0002.pdf).** Figures S5 to S7.
**Table S1 (mSystems01435-25-s0003.xlsx).** Sample read counts.
**Table S2 (mSystems01435-25-s0004.xlsx).** Summary of vaccine immune response results in this study.
**Table S3 (mSystems01435-25-s0005.xlsx).** Taxonomic assignment of ASVs.
**Table S4 (mSystems01435-25-s0006.xlsx).** PCR primers for virome sequencing.
**Table S5 (mSystems01435-25-s0007.xlsx).** Microbiome comparison analysis between different geography regions.
**Table S6 (mSystems01435-25-s0008.xlsx).** Summary of bacterial ASVs associated with HIV clade and antigen specific immune response within Rwanda and U.S. region.

### Open Peer Review

**PEER REVIEW HISTORY (review-history.pdf).** An accounting of the reviewer comments and feedback.

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
