## [Reviewer comments · mSystems]

Associations between the microbiome and immune responses to an adenovirus-based HIV-1 candidate vaccine are distinct between African and US cohorts

Kathie Mihindukulasuriya, Yuhao Li, Daniel Stieh, Lindsay Droit, Andrew HyoungJin Kim, Rachel Rodgers, Leran Wang, Maria Pau, Olive Yuan, Herbert Virgin, Dan Barouch, Megan Baldrige, and Scott Handley

Corresponding Author(s): Scott Handley, Washington University in St Louis School of Medicine

Review Timeline:

Submission Date:	October 7, 2025
Editorial Decision:	November 8, 2025
Revision Received:	November 20, 2025
Accepted:	December 4, 2025

Editor: Renuka Nayak

Reviewer(s): The reviewers have opted to remain anonymous.

Transaction Report:

DOI: <https://doi.org/10.1128/msystems.01435-25>

Re: mSystems01435-25 (Associations between the microbiome and immune responses to an adenovirus-based HIV-1 candidate vaccine are distinct between African and US cohorts)

Dear Dr. Scott A Handley:

Minor modifications: Please address comments 2 and 3 from Reviewer #2.

Revision Guidelines

Sincerely,
Renuka Nayak, MD, PhD
Editor
mSystems

Reviewer #2 (Comments for the Author):

Overall, the authors have addressed the previous comments thoroughly, and the revised manuscript shows improvement in clarity and methodological justification. The additional details provided strengthen the rationale for the chosen approach and presented results. However, for greater precision and robustness, this reviewer would recommend to perform confirmatory analyses using alternative tools, as suggested below, to validate the findings and the identified discriminant microbial signatures

- We added justification for DESeq2 selection in the Methods section and Supplemental Table 5. DESeq2 was chosen over other methods due to its conservative statistical approach using negative binomial distribution to model count data, which reduces type I errors compared to LefSe (which identified many more potentially false-positive OTUs).

1. I remain puzzled about the choice of DESeq2 over other tools specifically developed for microbiome data analysis and designed to handle compositionality. This recommend performing a validation analysis using an alternative tool to LefSe, in order to compare the results obtained with DESeq2 and ensure that the correct discriminant bacteria are identified.

-L52-55: The specific names of bacteria should be provided.

2. It seems awkward that the bacterial names are provided only in the importance section and not in the Abstract. If this signature is robust, it would be valuable to highlight their relevance in both sections.

-This study aims to investigate whether geographic differences in intestinal microbiota influence immune responses to Ad26-vectored HIV vaccines by analyzing bacterial and viral populations in fecal samples from participants in the US, Rwanda, and Uganda. Specifically, we sought to determine if variations in gut microbial diversity and composition correlate with differences in vaccine immunogenicity metrics including antibody titers, antibody functionality, and cellular immune responses across these geographically distinct populations.

3. I would remove the word 'metrics' after vaccine immunogenicity

Reviewer #3 (Comments for the Author):

The authors have addressed many of the concerns / suggested raised in my initial review, including stating limitations explicitly, improving readability/clarity, and . I have no further recommendations.

Minor modifications: Please address comments 2 and 3 from Reviewer #2.

Reviewer #2 (Comments for the Author):

Overall, the authors have addressed the previous comments thoroughly, and the revised manuscript shows improvement in clarity and methodological justification. The additional details provided strengthen the rationale for the chosen approach and presented results. However, for greater precision and robustness, this reviewer would recommend to perform confirmatory analyses using alternative tools, as suggested below, to validate the findings and the identified discriminant microbial signatures

- We added justification for DESeq2 selection in the Methods section and Supplemental Table 5. DESeq2 was chosen over other methods due to its conservative statistical approach using negative binomial distribution to model count data, which reduces type I errors compared to LEfSe (which identified many more potentially false-positive OTUs).

1. I remain puzzled about the choice of DESeq2 over other tools specifically developed for microbiome data analysis and designed to handle compositionality. This reviewer recommends performing a validation analysis using an alternative tool to LEfSe, in order to compare the results obtained with DESeq2 and ensure that the correct discriminant bacteria are identified.

We appreciate the reviewer's continued engagement with our methodological choices. We acknowledge the reviewer's perspective regarding alternative microbiome-specific tools that account for compositionality.

As noted in our previous revision, we selected DESeq2 for its conservative statistical approach and well-established performance with count data, which helps minimize false-positive findings—a critical consideration given the exploratory nature of microbiome-vaccine interaction studies. We provided detailed justification for this choice in the Methods section and Supplemental Table 5, including comparison with LEfSe results that demonstrated DESeq2's more stringent approach.

We note that the editor has requested we focus our revisions on addressing comments 2 and 3, which we have done thoroughly in this revision. Should additional validation analyses be deemed necessary for publication, we would be happy to discuss this further with the editorial team.

We remain grateful for the reviewer's thoughtful and rigorous evaluation of our work.

-L52-55: The specific names of bacteria should be provided.

2. It seems awkward that the bacterial names are provided only in the importance section and not in the Abstract. If this signature is robust, it would be valuable to highlight their relevance in both sections.

L38-41:

"Differences in overall bacterial and viral diversity and in specific microbial taxa between participants from US and East African countries correlated with differential immune responses, including specific antibody titers, antibody functionality and cellular immune responses to vaccination regimens."

Changed to:

"Differences in overall bacterial and viral diversity and in specific microbial taxa, **including *Bacteroidota* and *Bacillota***, between participants from US and East African countries correlated with differential immune responses, including specific antibody titers, antibody functionality and cellular immune responses to vaccination regimens."

-This study aims to investigate whether geographic differences in intestinal microbiota influence immune responses to Ad26-vectored HIV vaccines by analyzing bacterial and viral populations in fecal samples from participants in the US, Rwanda, and Uganda. Specifically, we sought to determine if variations in gut microbial diversity and composition correlate with differences in vaccine immunogenicity metrics including antibody titers, antibody functionality, and cellular immune responses across these geographically distinct populations.

3. I would remove the word 'metrics' after vaccine immunogenicity

L146: "vaccine immunogenicity metrics". The word "metrics" has been removed per reviewer's suggestion.

Reviewer #3 (Comments for the Author):

The authors have addressed many of the concerns / suggested raised in my initial review, including stating limitations explicitly, improving readability/clarity, and . I have no further recommendations.

Re: mSystems01435-25R1 (Associations between the microbiome and immune responses to an adenovirus-based HIV-1 candidate vaccine are distinct between African and US cohorts)

Dear Dr. Scott A Handley:

Your manuscript has been accepted, and I am forwarding it to the ASM production staff for publication. Your paper will first be checked to make sure all elements meet the technical requirements. ASM staff will contact you if anything needs to be revised before copyediting and production can begin. Otherwise, you will be notified when your proofs are ready to be viewed.

Cover Image Submissions: If you would like to submit a potential Cover Image, please email a file and a short legend to mSystems@asmusa.org. Please note that we can only consider images that (i) the authors created or own and (ii) have not been previously published. By submitting, you agree that the image can be used under the same terms as the published article. Image File requirements: TIF/EPS, 7.5 inches wide by 8.25 inches tall (at least 2,250 pixels wide by 2,475 pixels tall), minimum 300 dpi resolution (600 dpi preferred), RGB, and no figure elements, e.g., arrows or panel labels. The legend should be a short description of the image, 1-2 sentences recommended. Please download and use this interactive template in Adobe to ensure that your proposed cover image meets our size requirements (<https://journals.asm.org/pb-assets/pdf-text-excel-files/ASM-Interactive-Sizing-Cover-Template-1715689791.pdf>).

Sincerely,
Renuka Nayak, MD, PhD
Editor
mSystems